# Fracturing Fluid Polymer Thickener with Superior Temperature, Salt and Shear Resistance Properties from the Synergistic Effect of Double-Tail Hydrophobic Monomer and Nonionic Polymerizable Surfactant

**DOI:** 10.3390/molecules28135104

**Published:** 2023-06-29

**Authors:** Shenglong Shi, Jinsheng Sun, Kaihe Lv, Jingping Liu, Yingrui Bai, Jintang Wang, Xianbin Huang, Jiafeng Jin, Jian Li

**Affiliations:** 1Department of Petroleum Engineering, China University of Petroleum (East China), Qingdao 266580, China; sunjsdri@cnpc.com.cn (J.S.); lkh54321@126.com (K.L.); liujingping20@126.com (J.L.); smart-byron@163.com (Y.B.); wangjintang@upc.edu.cn (J.W.); 20170092@upc.edu.cn (X.H.); jjf5211314@126.com (J.J.); cuplijian@sina.com (J.L.); 2CNPC Engineering Technology R&D Company Limited, Beijing 102206, China

**Keywords:** fracturing fluid polymer thickener, high-salinity and high-temperature, hydrophobic association, linear entanglement, synergistic effect

## Abstract

To develop high-salinity, high-temperature reservoirs, two hydrophobically associating polymers as fracturing fluid thickener were respectively synthesized through aqueous solution polymerization with acrylamide (AM), acrylic acid (AA), 2-acrylamido-2-methylpropanesulfonic acid (AMPS), nonionic polymerizable surfactant (NPS) and double-tail hydrophobic monomer (DHM). The thickener ASDM (AM/AA/AMPS/NPS/DHM) and thickener ASD (AM/AA/AMPS/DHM) were compared in terms of properties of water dissolution, thickening ability, rheological behavior and sand-carrying. The results showed that ASDM could be quickly diluted in water within 6 min, 66.7% less than that of ASD. ASDM exhibited salt-thickening performance, and the apparent viscosity of 0.5 wt% ASDM reached 175.9 mPa·s in 100,000 mg/L brine, 100.6% higher than that of ASD. The viscosity of 0.5 wt% ASDM was 85.9 mPa·s after shearing for 120 min at 120 °C and at 170 s^−1^, 46.6% higher than that of ASD. ASDM exhibited better performance in thickening ability, viscoelasticity, shear recovery, thixotropy and sand-carrying than ASD. The synergistic effect of hydrophobic association and linear entanglement greatly enhancing the performance of ASDM and the compactness of the spatial network structure of the ASDM was enhanced. In general, ASDM exhibited great potential for application in extreme environmental conditions with high salt and high temperatures.

## 1. Introduction

The declining number of conventional oil and gas fields has made unconventional oil and gas reservoirs with ultra-low permeability increasingly important to the world’s energy supply. Hydraulic fracturing is important to effectively open reservoir pore-throat spaces, and therefore affects the economic and environmental outcomes of oil fields. Hydraulic fracturing is a typical method to increase hydrocarbon production by creating highly conductive fracture networks around wellbores [1]. During the hydraulic fracturing process, fracturing fluids are crucial to fracture production, pressure transmission, and distribution of proppants within the fractures [2]. The fracturing fluid influences the success or failure of hydraulic fracturing operations in a given area, as well as the degree of production improvement.

In general, water-based fracturing fluids such as guar gum, synthetic polymers and viscoelastic surfactants are the most commonly used types [3,4]. Nevertheless, guar gum lacks thermal stability and produces a large amount of residue upon breaking, which has a negative impact on productivity [5,6]. High filtration loss, high cost, and difficulty in breaking micelles have limited the widespread use of viscoelastic surfactant thickeners, which rely on molecular self-assembly to create worm-like micelles to deliver proppants into fractures [7,8]. Water-soluble polymers with a small number of hydrophobic groups on the backbone are known as hydrophobically associating water-soluble polymers (HAWSP). Because of the introduction of small amounts of hydrophobic monomers, these polymer molecules aggregate and form a unique spatial network structure that allows them to achieve excellent sand-carrying ability without the use of crosslinkers and also break down quickly, leaving little residue throughout the flow back to the surface [9,10,11].

However, the temperature and concentration of mineral salt ions increase dramatically in the harsh reservoir environment. As a result, charge-shielding effects and cross-linking action between ions and HAWSP occur simultaneously. Precipitation will occur at high-salinity brine and high temperatures, causing several limitations to practical applications of HAWSP [12,13,14]. In order to solve these problems, many researchers have introduced functional monomers in recent years to improve the salt resistance and temperature resistance of hydrophobically associating polymers in oil and gas field development [15,16,17,18]. 2-acrylamide-2-methylpropanesulfonic acid (AMPS) [19], 4-isopropenylcarbamoyl-benzene sulfonic acid (AMBS) [20], N-vinyl-2-pyrrolidone (NVP) [21], Acryloyl morpholine (ACMO) [22], nano-silica [23], and large rigid groups were used to synthesize salt-resistant and temperature-resistant hydrophobic-association polymers [24]. Some researchers have used two-tailed monomers to synthesize hydrophobic-related polymers with greatly enhanced salt and temperature resistance compared to single-tailed hydrophobic-related polymers [25].

Currently, the common method of preparing HAWSP is micellar copolymerization, whereby hydrophobic monomers are dissolved in water with the addition of an appropriate surfactant and then copolymerized with water-soluble monomers. However, HAWSP prepared through micellar copolymerization usually have some disadvantages, i.e., the long dissolution time and slow dissolution rate of HAWSP; unreacted residual surfactants may lead to environmental problems, and the presence of surfactants in the final product may cause some adverse effects on product performance [26,27,28,29]. The introduction of polymerizable surfactant can solve these problems in traditional surfactant applications. Polymerizable surfactant molecules incorporate polymerizable reactive groups, mainly unsaturated alkenes, into the structure of conventional surfactant molecules. In contrast to conventional surfactants, polymerizable surfactants not only possess the characteristics of conventional surfactants, but also can be copolymerized with unsaturated monomers at high temperature or in the presence of initiators, and these compounds can also be used to improve the solubility and association of HAWSP [30,31,32]. Polymerizable surfactants can replace common hydrophobic monomers and surfactants and can be further locked into these new supramolecular structures by polymerization [33,34,35].

Mao et al. [36] synthesized a hydrophobically-associating polymer (SRHV) by using acrylamide, acrylic acid, 2-acrylamide-2-methylpropanesulfonic acid, the cationic long-chain hydrophobic monomer, and nonionic polymerizable surfactant. SRHV can achieve low-viscosity and high-elasticity rheological behaviors in high-salinity brine environments by constructing a strong hydrophobic association network, thereby enhancing sand-carrying performance. Zhang et al. [37] developed a hydrophobically-associating polymer, PDMA, with double-tail polymerizable surfactants and long-chain hydrophobic monomer that is resistant to high-salinity formation water. PDMA exhibited good thixotropy, good viscoelastic behavior and shear resistance at high salt concentrations. Fan et al. [38] prepared a modified polymer, MHAP, with acrylamide, acrylic acid, 2-acrylamide-2-methylpropanesulfonic acid, the cationic hydrophobic monomer, and polymerizable surfactant with oxyethylene groups (-(CH_2_-CH_2_-O)_n_-). MHAP demonstrated good thickening property, rheological property, temperature-resistance sensitivity, and salt resistance/sensitivity stimulation response. Although considerable research has been done on the effects of temperature and salt ions on HAWSP synthesized from cationic hydrophobic monomers and polymerizable surfactants, and both cationic hydrophobic monomer and polymerizable surfactants are beneficial in improving the temperature resistance, salt resistance and shear resistance of hydrophobically associating polymers, the synergistic effects of cationic hydrophobic monomers and polymerizable surfactants on the performance of HAWSP have rarely been considered in relevant studies.

In this study, a novel hydrophobically associating polymer (ASDM) was synthesized by the homogeneous-phase aqueous copolymerization of acrylamide (AM), acrylic acid (AA), and 2-acrylamido- 2-methylpropane sulfonic acid (AMPS), nonionic polymerizable surfactant (NPS) and double-tail hydrophobic monomers (DHM) using a complex initiation system, which contained ammonium persulfate/sodium hydrogen sulfite/ascorbic acid/water-soluble azo; all monomer structures and the chemical composition of the polymers are shown in Figure 1. Double-tail hydrophobic monomers were grafted onto the polymer to increase the polymer-to-polymer association and further enhance the good features of fracturing fluid, such as temperature resistance and shear resistance. The nonionic polymerizable surfactant with oxyethylene groups embedded in the molecules showed good water solubility, and the repulsion between the hydrophobic chain and oxyethylene group was more favorable for the extension of polymer, which obtained high viscosity and shear resistance in high-salinity brine [39]. The critical association concentration, dissolution rate, salt tolerance, rheological behavior, shear recovery capacity, sand-carrying performance, temperature- and shear-resistance, and microstructure of ASDM were investigated; the synergistic effects of double-tail hydrophobic monomers and nonionic polymerizable surfactants on the performance of ASDM were also discussed.

## 2. Results and Discussion

### 2.1. Characterization of Thickeners

Figure 2 shows the FTIR spectra of polymers ASD and ASDM. As could be seen in Figure 1A, the peak at 3428 cm^−1^ was the stretching vibration of the amide bond N-H. The peak at 1636 cm^−1^ was the bending vibration absorption peak of C=O, which validated the presence of acrylamide in the polymer. The stretching vibration peaks at 2936 cm^−1^ and 2871 cm^−1^ were assigned to stretching vibration absorption associated with -CH_2_- and -CH_3_; the peak at 1106 cm^−1^ was the stretching vibration of the ethoxy ether bond -C-O-C- of DMAOZ2, which proved that DHM was inserted into the framework of the polymer. The peak at 1198 cm^−1^ was the asymmetric stretching vibration of the S=O bond and the peak at 615 cm^−1^ was the stretching vibration peak of the C-S in the sulfonate, which verified the successful embedment of AMPS into the polymer backbone. The peak at 1557 cm^−1^ was an antisymmetric contraction of the multi-electron conjugate system of the carboxylate, while the peak at 1406 cm^−1^ marked a symmetric contraction of the carboxylate, which showed the attachment of acrylic acid to the polymer backbone. The structure of the polymer ASD corresponded to the designed structure of the target polymer. A strong absorption peak at 1032 cm^−1^ was observed in Figure 1B, corresponding to C-O-C (second alcohol ester) stretching vibrations of NPS [37], which indicated that the NPS monomer reacted with ASD to form the expected product, ASDM. As the groups in the molecular structures of the two polymers were basically similar, the positions of the peaks in the infrared spectra of the two polymers were basically the same, but the intensity of the peaks of ASDM was weaker than those of ASD, indicating that the nonionic polymerizable surfactant NPS had a certain weakening effect on the intensity of the absorption peaks of the other groups, and also indicating that NPS was involved in the polymerization reaction.

Figure 3 shows the ^1^H NMR (400 MHz, D_2_O) spectrum of ASDM. It was shown from Figure 3 that the proton signal at 4.70 ppm was allocated to the solvent proton (D_2_O). 0.78 ppm (a, -CH_3_), 1.10–1.12 ppm (b, -CH_2_-CH_3_), 1.22 ppm (b, -CH_2_-CH_3_), 1.44 ppm (d, -CH_3_, AMPS), 1.56 ppm (m, -CH_2_-), 2.12 ppm (e, -CH-), 2.41 ppm (l, -CH_2_-COO-), 3.27 ppm (c, -CH_2_-CH_2_-), 3.55 ppm (g, -CH_3_), 3.57 ppm (h, -CHCH_2_-N-), 3.59 ppm (i, -CH-CH_2_-SO_3_), 3.60 ppm (k, -C-CH_2_-SO_3_, AMPS), 3.63 ppm (f, -O-CH_2_-), 5.56–5.59 ppm (j, -O-CH-). Therefore, ^1^H NMR analyses confirmed that the polymer produced in this study was largely consistent with the designed polymer, which indicated successful synthesis.

### 2.2. Critical Association Concentration

The critical association concentration (CAC) of hydrophobically associated polymers can be used as a threshold to describe the behavior of the polymer. Figure 4 shows the curves of viscosity versus polymer concentration. It is easy to see that there are two distinct shifts in ASDM at concentrations of 0.086 wt% and 0.193 wt%, and the ASDM curve was divided into three regions by these two points. These two points were named as the first critical association concentration (C_1_) and second critical association concentration (C_2_), respectively, in the following. When ASDM concentration was below C_1_, there were fewer polymer macromolecules in solution, thus inducing fewer intramolecular associated microregions, which were formed by hydrophobic interactions between the hydrophobic groups of the polymer solution. When ASDM concentration was between C_1_ and C_2_, the polymer molecules formed a supramolecular structure that was on the basis of intermolecular bonding-“dynamic network structure” and the increasing hydrodynamic size increased the viscosity of the solution; the viscosity of the polymer solution increased with increasing ASDM concentration. When ASDM concentration was greater than C_2_, the intermolecular entanglement and intermolecular association became stronger, thereby inducing a multilayer spatial network structure in aqueous solution and enhancing the density of the network structure in solution [36]; therefore, there was a marked increase in the viscosity of the polymer solution when ASDM concentration exceeded C_2_.

There was only one obvious change point for ASD, as shown in Figure 4. In addition, the viscosity of the ASDM solution was higher than that of the ASD solution for the same polymer concentration. This was due to the special structure of ASDM, which contained the hydrophobic long chains and oxyethylene groups, whereas ASD only possessed hydrophobic long chains. The length of the NPS hydrophobic chain was similar to that of the DHM hydrophobic chain. The association behavior of polymer ASDM in water could be generated automatically by the “entropy-driven process” to form dynamic, reversible physically crosslinked aggregates. The modified supramolecular polymer ASDM was considered to increase the number and structural strength of hydrophobic bonds and improved the molecular weight and hydrodynamic radius of the polymer in water due to the synergistic effects of the hydrophobic association network of DHM and linear entanglement of NPS existed simultaneously between polymers. Hence, a dense interlaced network structure, high interface adhesion and a highly viscoelastic fluid were achieved. Therefore, the viscosity of ASDM was greater than that of the ASD solution at the same polymer concentration [38].

### 2.3. Dissolution Rate

In general, as the polymer is dissolved gradually and stretched in water, the conductivity of the solution increases gradually. The solution’s conductivity then remains constant after the polymer has been stretched completely. The dissolution rate of polymer can therefore be reflected by the rate of increase in the solution’s conductivity. Figure 5 shows the change in conductivity of solution during the dissolution of ASD and ASDM. After the polymer was added to the water, the solution’s conductivity gradually increased with time. Finally, the solution’s conductivity remained unchanged, indicating that the polymer had been completely dissolved. The final dissolution times in water for ASD and ASDM were 10 min and 6 min, respectively; this result showed that these two polymers revealed good solubility. In addition, the dissolution rate of ASDM was better than that of ASD, showing that monomer NPS in polymer ASDM could enhance the dissolution of the polymer, which was mainly attributed to the strong water solubility of oxyethylene groups in NPS. As we know, hydrogen bonds could be formed between oxygen atoms in the oxyethylene groups and water molecules, which enhanced the solubility of the polymer molecular chains and promoted the entry of water molecules into the polymer particles [40]. As a result, the polymer particles gradually swelled, and tightly entwined polymer molecular chains separated from each other, leading to rapid dissolution of polymers. In addition, the repulsive forces of hydrophilic and hydrophobic segments of the polymer chains could promote the chain extension of polymers. [41] Although the presence of long hydrophobic chains had a negative impact on the water solubility of polymers, the polymers could be dissolved in less than 7 min, exhibiting rapid water solubility. The rapid-dissolving property of ASDM facilitated its preparation and application in hydraulic fracturing.

### 2.4. Salt Resistance

The effects of salt on the viscosity of the polymer solution by measuring the viscosity at different concentrations of compound salt brine (NaCl and CaCl_2_ at a mass ratio of 10:1), MgCl_2_ and CaCl_2_ are shown in Figure 6. As shown in Figure 6A, the viscosity of the 0.5 wt% ASD decreased as the salt concentration increased. At high salt concentration, hydrophobic monomers reduced the solubility of the polymer, resulting in a rapid decrease in its viscosity. The viscosity of 0.5 wt% ASD was below 90 mPa·s when the salt concentration was greater than 10 × 10^4^ mg/L; the viscosity of the ASDM solution was increased significantly with the introduction of NPS compared to ASD; the viscosity of 0.5 wt% ASDM solution increased from 162.1 mPa·s to 175.9 mPa·s when salt concentration reached 10 × 10^4^ mg/L, 100.6% higher than that of ASD, and the ASDM exhibited salt thickening. As the salt concentration increased further, the viscosity of the ASDM solution decreased gradually, and finally decreased to a stable value of about 118.6 mPa·s. When the concentration of compound salt reached 200,000 mg/L, the viscosity of HPAM solution decreased almost to zero.

The polymer chains usually undergo curling and viscosity loss due to the electric double-layer compression of the polymer hydration shell and electrostatic shielding effect induced by metal ions [42]. For hydrophobically associating polymer ASDM, AMPS was widely used as a salt-tolerant functional group in polymer design because of its strong anionic and water-soluble sulfonic acid group, which was not sensitive to attack by external ions [24]. The introduction of oxyethylene groups in NPS with strong hydration capacity improved the solubility and inhibited hydrolysis of the ASDM at high brine concentrations. In addition, the hydrophobic association effect between NPS and DHM was enhanced due to solution polarity enhancement; the electric double layer was compressed by salt ions, which facilitated the association of hydrophobic chains. Complexation reactions would occur between multivalent metal ions and oxyethylene groups by filling the unoccupied orbital of metal ions through the lone pair of electrons from the oxygen of the oxyethylene groups. As a result, the repulsion between hydrophobic groups and oxyethylene groups was enhanced, subsequently promoting the stretching of the polymer chains. All of the above factors increased the hydrodynamic volume of polymers and the viscosity of the polymer solution, contributing to the resistance of inorganic mineral salts. Therefore, an increase in salt concentration had a positive effect on the viscosity of the ASDM solution. Subsequently, more inorganic salts were added to the polymer solution, and the excess inorganic salt ions dehydrated the polymer molecules; the electrostatic shielding effect of salt on polymer molecular chains exceeded the hydrophobic association effect between polymer molecules, leading to a thinning of the hydration shells of the polymer molecules. Finally, the viscosity of the polymer solution decreased at high salt concentrations. Meanwhile, the carboxylate ions on the HPAM molecule were shielded by the metal cation in the brine, causing the polymer chains to curl, and the viscosity of the HPAM solution decreased sharply.

The effect of MgCl_2_ and CaCl_2_ on the viscosities of polymer solutions were also investigated, and the results are shown in Figure 6B. With the increase in the MgCl_2_ and CaCl_2_ solution concentration, the viscosities of the polymer solutions decreased. Obviously, the viscosity of the ASDM solution was greater than that of the ASD solution with increased MgCl_2_ and CaCl_2_, indicating a synergistic effect between NPS and DHM. When the concentrations of MgCl_2_ and CaCl_2_ reached 2 × 10^4^ mg/L, the viscosities of ASDM solutions decreased from 106.5 mPa·s to 74.3 mPa·s and 83.2 mPa·s, respectively. However, the viscosity of the HPAM solution was almost zero when the concentrations of calcium chloride and magnesium chloride reached 7000 mg/L, indicating that the polymer ASDM obtained excellent salt tolerance compared with HPAM. In addition, it can be seen from Figure 5 that the three metal ions of Na^+^, Ca^2+^, and Mg^2+^ had different effects on the viscosity of the polymer solution, which was caused by the different hydration ionic radiuses of the three metal ions. The larger the radius of hydrated ions, the greater the attraction between the salt ions and the water molecules, and the stronger the dehydration ability of inorganic salt ions. From a macroscopic point of view, the salt ions (Ca^2+^ and Mg^2+^) with a large hydrated ionic radius caused the viscosity of the polymer solution to decrease sharply to be stable. The influence mechanism of divalent ions was the same as that of monovalent ions, but the compression of the electric double layer of the polymer hydration shell was strengthened.

### 2.5. Viscoelasticity

Figure 7A–D shows the relation curves between the storage modulus (G′), loss modulus (G″), and strain of polymer solutions with different concentrations of ASD and ASDM. For ASD, when the concentration was 0.05 wt%, G′ was lower than G″, indicating that the polymer was primarily viscous. However, at a concentration of 0.1 wt%, G′ was greater than G″ when the strain was less than 6%, indicative of elastic fluids. For ASDM, G′ was higher than G″ under different polymer concentrations, indicating that the ASDM was mainly elastomeric. With the increase in shear strain, the linear-plateau region of ASDM was longer than that of ASD, implying that the ASDM molecules obtained a higher association stability. Thus, a high shear strain had no significant effect on the associative structure of ASDM, but it markedly influenced the plateau region of ASD.

Figure 7E–H shows the relationships between the storage modulus and loss modulus versus frequency. During the frequency sweep of the test range, the changes were similar for all the test solutions, with G′ > G″ for ASDM. The addition of NPS monomer increased both G′ and G″. This might be due to the higher degree of association in the ASDM solution compared to the ASD solution, resulting in the ASDM molecules forming a compact 3D web-like structure, reflecting the bearing of more physical entanglements. The larger aggregates could not exhibit rapid relaxation at high frequencies due to the presence of external forces, leading to the storage of elastic energy. Therefore, the elasticity increased with the frequency, indicating denser hydrophobic domains, stronger intermolecular interactions, and higher structural stability of the synthesized ASDM polymer [43].

### 2.6. Thixotropy

Thixotropy is a reversible sol–gel phenomenon that appears in polymer solutions and represents the time dependence of fluid viscosity. The larger the area of closure loop, the better thixotropy of the polymer solution [44]. The thixotropic loops of the 0.5 wt% ASDM solution and 0.5 wt% ASD solution are shown in Figure 8. As the shear rate increased, the shear stress in the polymer solution increased gradually, indicating a gradual increase in the energy required to break the solution. By comparing the thixotropic rings of the ASDM solution and ASD solution, it was found that the thixotropic loop area of the ASDM solution was larger than that of the ASD solution, and the maximum shear stress required for ASDM solution and ASD solution was 19.4 Pa and 10.5 Pa, respectively, indicating that the energy required to destroy the ASDM solution was larger than that of the ASD solution, enabling ASDM to exhibit a better thixotropy compared to ASD. This confirmed that oxyethylene groups could improve the thixotropy of the ASDM solution, increasing the structural strength of the polymer in the solution; the synergistic effect of rigid hydrophobic association and linear entanglement was enhanced and the compactness of the spatial network structure of the polymer was enhanced because of the strong hydrophilicity of oxyethylene groups. As a result, the shear stress required to destroy the ASDM solution increased.

### 2.7. Shear Recovery

The shear recovery performance of ASDM and ASD determines the actual effect of the fracturing fluid in the formation after shearing in injection, and the experimental results are shown in Figure 9. The shear recovery rate of 0.5 wt% ASDM and 0.5 wt% ASD reached 99.6% and 94.5%, respectively. These results showed that both ASDM and ASD had good shear recovery properties. It was believed that this phenomenon was due to the reversible spatial network structure formed by the polymer in hydrophobic association. The hydrophobic association structure was disrupted at high shear rates, leading to a rapid decrease in the viscosity of the polymer solution. When the shearing effect was reduced, the physical crosslinking between the polymer molecular chains was reformed and the viscosity was restored. The ASDM and ASD solution had good shear viscosity recovery properties because of the unique reversibility of their associative structures. The shear recovery rate of ASDM was higher than that of ASD, which could be attributed to the presence of long-chain oxyethylene groups and double-tailed hydrophobic groups; the strong hydrophilic force interactions among the chains that surrounded the oxyethylene groups made the molecular chains stretch, and the alkyl chains reinforced the hydrophobic association effect between polymer molecules [44].

### 2.8. Sand-Carrying Performance

The sand-carrying performance of ASDM solutions and ASD solution at various salt concentrations and different temperatures were investigated and compared; the results are shown in Table 1. The sand-carrying capacity of ASDM was much better than that of ASD. When the ASDM concentration increased from 0.3 to 1.0 wt% and the temperature increased from 25 to 140 °C, the settling velocities were 0.028, 0.082, 0.153, and 0.228 mm/s, respectively, showing that the ASDM exhibited good sand-carrying performance even at high temperatures. These results showed that the modification of polymer with hydrophobic groups could enhance the structural bonding and viscoelasticity of the spatial network, thus achieving viscoelastic “stacking” and proppant-carrying. At the same time, hydrophobic groups of the polymer could combine with nonionic polymerizable surfactant to re-strengthen the viscoelasticity of the system and achieve characteristics of high sand-carrying and deep sand-spreading. This allowed the goal of improving the reservoir’s seepage characteristics to be achieved.

### 2.9. Temperature and Shear Resistance Performance

The high-temperature shear test curves of ASDM and ASD are shown in Figure 10. As the temperature increased, the viscosity gradually decreased and then flattened out, demonstrating that dynamic linear entanglement and hydrophobic association were continuously generated under continuous shearing. The viscosities of 0.5 wt% ASD and 0.5 wt% ASDM were 58.6 and 85.9 mPa·s after a constant shear of 170 s^−1^ at 120 °C for 120 min. Compared with ASM solution, the viscosity of the ASDM solution increased significantly, which might be related to the thermal motion of polymer molecules, indicating that modification with NPS monomer enhanced the temperature resistance of polymers significantly. On the one hand, hydrophobically associated micelles became large due to the repulsive effect of long-chain hydrophobic groups linked to the oxyethylene groups. On the other hand, the presence of oxyethylene groups caused an increase in linear entanglement, resulting in a denser structure of the spatial chain entanglement network structure. Therefore, the introduction of oxyethylene groups in the polymer facilitated the formation of complex spatial networks through linear entanglement and hydrophobic association, contributing to the increase in viscosity of ASDM solutions. It was worth noting that the increase in spatial network structure at high polymer concentrations was more conducive to the formation of dynamically cross-linked intermolecular linear entanglement and hydrophobically associating structures; thus, a large-scale supramolecular structure and better temperature and shear resistance performance was obtained. Thus, ASDM showed great potential for practical applications in thermal environments.

### 2.10. Microscopic Morphology and Mechanism Analysis

Figure 11 shows the SEM images of aqueous ASD and ASDM solutions. The ASD molecular chains were interwoven to form a tight three-dimensional network structure, crosslinking points and protrusions were observed at the nodes of the molecular chains of the ASD surface structure, which might be due to the hydrophobic-association effect exhibited by the hydrophobic functional groups within the ASD molecular chains. After the introduction of NPS, a strong compact spatial network structure of ASDM was formed between molecules, providing a dual structure of hydrophobic association and linear entanglement. For ASDM, the linear entanglement between some oxyethylene groups and the hydrophobically associated cationic structure could be observed clearly under microscopic images. As could be seen from Figure 11, when the monomer NPS was introduced, the number of hydrophobic groups increased and the association between hydrophobic chains was enhanced, making it easier to form supramolecular structures and resulting in a significant increase in the hydrodynamic volume of the polymer ASDM. Secondly, the oxyethylene groups of NPS complexed with metal salt ions and the lone pair of electrons of the oxygen atom on the oxyethylene groups were filled into the empty orbitals by the salt ions. In brine, the polymolecular aggregation promoted the formation of polymer solution through complexation, thereby increasing salt resistance and enhancing the polarity of the groups and the hydrophobicity of the polyether. In addition, the bridging effect of the long double-tail hydrophobic chains on the polymer backbone was similar to the support of a bridge abutment on the bridge deck, resulting in increased rigidity and thermal stability of the polymer chains. The synergistic effect of hydrophobic association and linear entanglement greatly enhances the performance of polymers, so ASDM exhibited the better temperature/salt/shear resistance stimulation response. Samples a and b in Figure 10 correspond to the gel states of the two polymers after they have been erected and then flattened. Meanwhile, the ASDM gel sample showed low slippage, indicating a tight intermolecular structure, while the ASD gel sample showed high slippage, indicating a loose intermolecular structure. This observation was consistent with the experimental results.

## 3. Materials and Methods

### 3.1. Materials

Acrylamide (AM, AR 99%), acrylic acid (AA, AR 99%), 2-acrylamido- 2-methyl propane sulfonic acid (AMPS, AR 98%), 2,2′-azobis (2-methylpropionamide) dihydrochloride (V_50_, AR 98%), ammonium persulfate (APS, AR 98%), sodium bisulfite (NaHSO_3_, AR 98%), ascorbic acid (V_C_, AR 98%), lauryl sodium sulfate (SDS, AR 98%), sodium hydroxide (NaOH, AR 98%), ethylenediaminetetraacetic acid disodium salt (EDTA-2Na, AR 98%), sodium chloride (NaCl, AR 99.5%), magnesium chloride (MgCl_2_, AR 98%), calcium chloride (CaCl_2_, AR 98%) and potassium bromide (AR 99%) were purchased from Aladdin BioChem Technology Co., Ltd (Shanghai, China). Nonionic polymerizable surfactant octadecyl polyoxyethylene ether methacrylate (NPS, industrial grade, 60%) was bought from Zhang Jiagang Render Chemicals Co., Ltd (Zhang Jiagang, China). Double-tail hydrophobic monomer (DHM) was synthesized in the laboratory. Proppant ceramsite with an apparent density of 2.45 g/cm^3^ and a size of 40/70 mesh, supplied by Jingang New Materials Co., Ltd. HPAM (*M*_w_ = 16~18 × 106), was purchased from the Beijign Hengju Oil Field Chemical Reagents Co., Ltd (BeiJing, China). Deionized (DI) water was obtained from a water purification system, compound salt brine was prepared with sodium chloride, and calcium chloride at a mass ratio of 10:1 was prepared in the lab. All chemicals were utilized without further purification.

### 3.2. Synthesis

#### 3.2.1. Preparation of Double-Tail Hydrophobic Monomer (DHM)

DHM was synthesized following a previously published method [45] (Figure 12).

#### 3.2.2. Synthesis of Thickeners ASD and ASDM

The reaction was carried out in a 500-mL beaker. The molar ratios of the monomers were adjusted to 2.484 mol/L of AM, 0.988 mol/L of AA, 0.095 mol/L of AMPS and 0.012 mol/L of DHM, and the total weight of the reacting mixture was 200 g by adding appropriate amounts of distilled water and 3 g of the surfactant SDS. The pH was adjusted to 7.0 with a 30 wt% NaOH solution and the reaction system was stirred at 10 °C for 60 min under a nitrogen atmosphere. Subsequently, an appropriate amount of initiator solution (based on 0.10% of the total mass of the monomer) was added through a syringe. The mass ratio of the components of the initiator was m(APS):m(NaHSO_3_):m(Vc):m(V_50_) = 11:5:6:2, which was kept under a thermal insulation system for 6 h. Subsequently, the gelatinous products were obtained and cut into small pieces, and the product was washed three times with ethanol and dried under vacuum at 70 °C for 48 h. The final product was ground into a fine powder. Figure 13 showed the synthetic route of ASD.

ASDM was synthesized using NPS as a polymerizable surfactant functional monomer. The reaction process was the same as that used for ASD, except that the SDS was removed and 0.025 mol/L of NPS was added (Figure 14).

### 3.3. Structural Characterization

Thickener powder and potassium bromide were placed in a mortar at a mass ratio of 1:75. After grinding and pressing, the sample was placed on a IRTracer-100 infrared spectrometer (SHIMADAZU, Japan) for transmission spectroscopy scanning. The spectrum was recorded in the range of 4000–700 cm^−1^ at 25 °C with a resolution of 4 cm^−1^. The nuclear magnetic resonance proton spectra (^1^H NMR) of the polymers in deuterium chloride (D_2_O) were measured using a Bruker AVANCE NEO 600 spectrometer (Bruker, Karlsruhe, Germany); the concentration of the polymer solution was 100 mg/L.

### 3.4. Critical Association Concentration

The viscosities of solutions containing 0.05–0.40 wt% thickener solutions were determined by using a HAAKE MARS 40 rheometer (HAAKE, Karlsruhe, Germany) with Cup Z43 cylinder plate (diameter = 43 mm) and CC41 rotor (diameter = 41 mm) at a shear rate of 170 s^−1^ and a temperature of 25 °C. The relationship curve between viscosity and concentration was obtained; the point at which a sudden change in viscosity occurred was considered to be the point of critical association concentration.

### 3.5. Dissolution Rate

The dissolution rate of the thickener was measured by the conductivity method. A solution’s conductivity was measured using a DDS-307+ conductivity meter (Chengdu Century Ark Technology Co., Ltd., Chengdu, China). The thickener samples were dispersed and dissolved in deionized water at 25 °C, and the changes in conductivity were investigated after the thickener had been added to the solution and completely dissolved.

### 3.6. Apparent Viscosity

The HAAKE MARS 40 rheometer with Cup Z43 cylinder plate and CC41 rotor was used for measuring the apparent viscosity of thickener at different salt concentration of 0–300,000 mg/L and at a temperature of 25 °C with a shear rate of 170 s^−1^.

### 3.7. Viscoelasticity

The thickener solutions were prepared with 100,000 mg/L synthetic brine, and the HAAKE MARS40 rheometer was used to measure the viscoelasticity of the ASDM solution. The cone plate geometry systems with cone PP35/Ti (diameter was 35 mm, gal was 1 mm) were selected for the measurement. The viscoelasticity was measured under the oscillatory shearing conditions at 25 °C. The strain amplitude ranged from 0.1 to 1000% and the oscillation frequency was 1 Hz. The strain values of thickener solution in the linear viscoelastic region were used to perform a frequency scanning of the solution to determine the viscoelastic strength in the range of 0.01–10 Hz.

### 3.8. Thixotropy

Thixotropy characterized the formation and destruction of thickener structures. The areas between the up-going and down-going curves were found to reflect this thixotropy. The thickener solutions were prepared with 100,000 mg/L synthetic brine; the thixotropy of thickener solutions was measured using a plate PP35/Ti by adjusting the three parameters in the rotary mode of HAAKE MARS 40. The experiment was divided into three stages: the shear rate in the first stage varied from 0.1 s^−1^ to 100 s^−1^, the second stage remained constant at 0 s^−1^ for 50 s, and the third stage was reduced from 100 s^−1^ to 0.1 s^−1^.

### 3.9. Shear Recovery Performance

The thickener solution was prepared with 100,000 mg/L synthetic brine, shearing the 0.5 wt% thickener solution using a HAAKE MARS 40 rheometer at various rates to simulate shear degradation. The experimental process was conducted as follows: (1) The shear rate was maintained at 40 s^−1^ for 3 min to obtain the initial viscosity; (2) the rate was increased to 1000 s^−1^ and shearing kept continuous for 2 min to ensure that the fluid was in a state of high speed shear; (3) the shear rate was adjusted back to 40 s^−1^ for 10 min to simulate the viscosity recovery process in the formation; (4) steps (2) and (3) were repeated twice. After these four steps, the shear-recovery behavior of the thickener was assessed.

### 3.10. Sand-Carrying Performance

The static sand-suspension experiments were carried out to test the sand-carrying performance of thickener by adding 40–70 mesh ceramsites at a 20% sand ratio at 25 °C, and the settling velocity of ceramsites was calculated.

### 3.11. Temperature and Shear Resistance Performance

A certain concentration of thickener solution was prepared with 100,000 mg/L synthetic brine, and the temperature- and shear-resistance performance of the thickener solution was determined using the HAKKE MARS40 rheometer in a temperature range of 25 °C–120 °C. The test was run for 120 min at a constant shear rate of 170 s^−1^. The rheometer utilized a high-pressure sealed concentric cylinder and rotor (PZ38 b), and the test required a sample volume of 32 mL.

### 3.12. Scanning Electron Microscopy

The microstructure of the aggregation pattern of thickener in aqueous solution was examined by scanning electron microscopy (SEM, Quanta 450, FEI, USA). All samples were dried at 25 °C and then frozen to −50 °C in liquid nitrogen. SEM was used to investigate the frozen surface of the samples at an accelerating voltage of 2.0 kV.

## 4. Conclusions

In order to promote temperature, salt and shear resistance of hydrophobically associating polymers as fracturing fluid thickener, two hydrophobically associating water-soluble polymers were respectively synthesized through aqueous solution polymerization with acrylamide (AM), acrylic acid (AA), 2-acrylamido-2-methylpropanesulfonic acid (AMPS), nonionic polymerizable surfactant (NPS) and double-tail hydrophobic monomer (DHM). The thickener ASDM (AM/AA/AMPS/NPS/DHM) and thickener ASD (AM/AA/AMPS/DHM) were compared by parameters such as water dissolution, thickening ability, rheological behavior, sand-carrying performance, and microstructure; the synergistic effects of DHM and NPS on performance of polymers were also discussed, and the main conclusions and recommendations were drawn as follows:(1)ASDM could be quickly diluted in water within 6 min, only two-thirds of the time required for dissolving ASD. ASDM exhibited salt-thickening performance, and the apparent viscosity of 0.5 wt% ASDM reached 175.9 mPa·s in 100,000 mg/L brine, 100.6% higher than that of ASD.(2)ASDM possessed better properties of thickening ability, viscoelasticity, thixotropy, sand-carrying and temperature and shear resistance than ASD due to the synergistic effect of hydrophobic association of DHM and linear entanglement of NPS.(3)The shear recovery rate of 0.5 wt% ASDM and 0.5 wt% ASD reached 99.6% and 94.5%; both ASDM and ASD had good shear viscosity recovery properties because of the unique reversibility of their associative structures.(4)The reasons why ASDM owned excellent temperature, salt and shear resistance properties were that a strong compact spatial network structure of ASDM was formed between molecules, providing a dual structure of hydrophobic association and linear entanglement. The bridging effect of the long double-tail hydrophobic chains DHM on the polymer backbone was similar to the support of a bridge abutment on the bridge deck, resulting in increased thermal stability and shear resistance of the polymer chains. The oxyethylene groups of NPS complexed with metal salt ions, and the lone pair of electrons of the oxygen atom on the oxyethylene groups were filled into the empty orbitals by the salt ions, increasing salt resistance and enhancing the polarity of the groups.

## Figures and Tables

**Figure 1 molecules-28-05104-f001:**
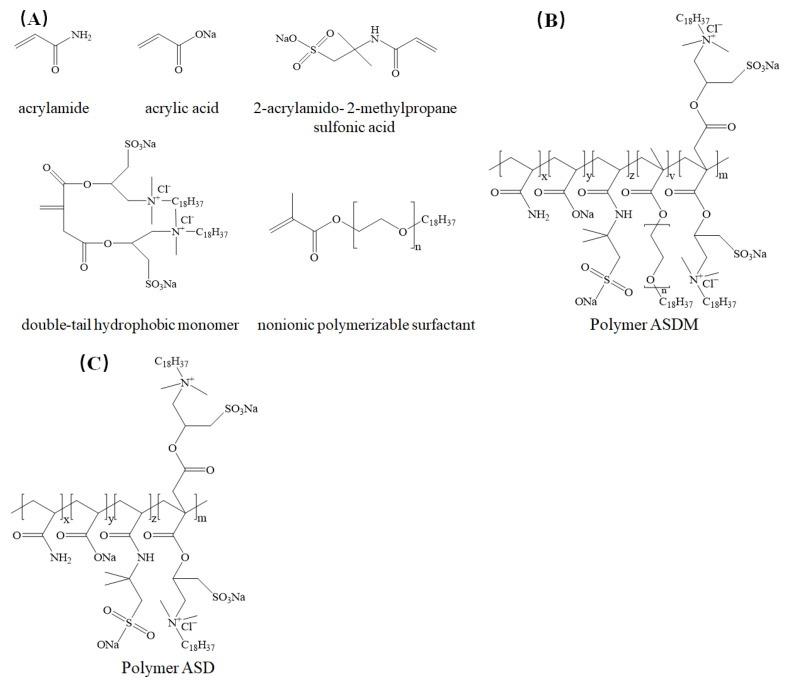
All monomer structures and chemical composition of the polymer ASDM. (**A**) Monomer structures. (**B**) Polymer ASDM. (**C**) Polymer ASD.

**Figure 2 molecules-28-05104-f002:**
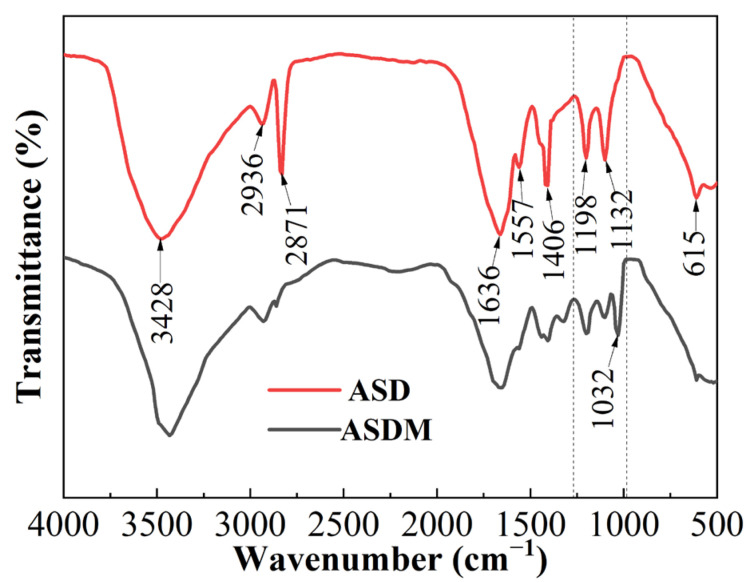
FTIR spectra of ASD and ASDM.

**Figure 3 molecules-28-05104-f003:**
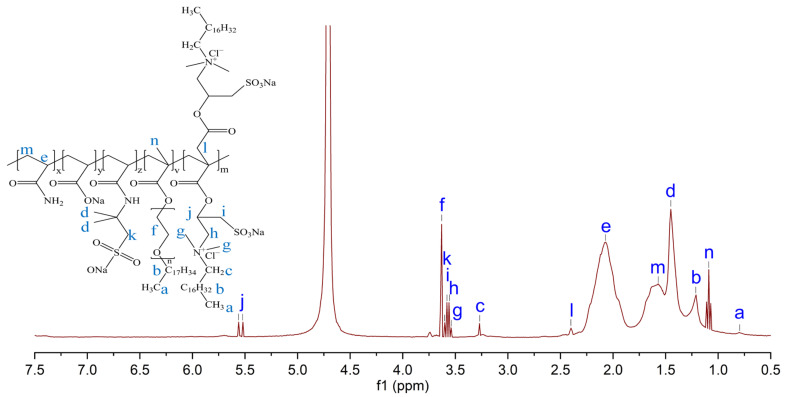
^1^H NMR spectra of ASDM.

**Figure 4 molecules-28-05104-f004:**
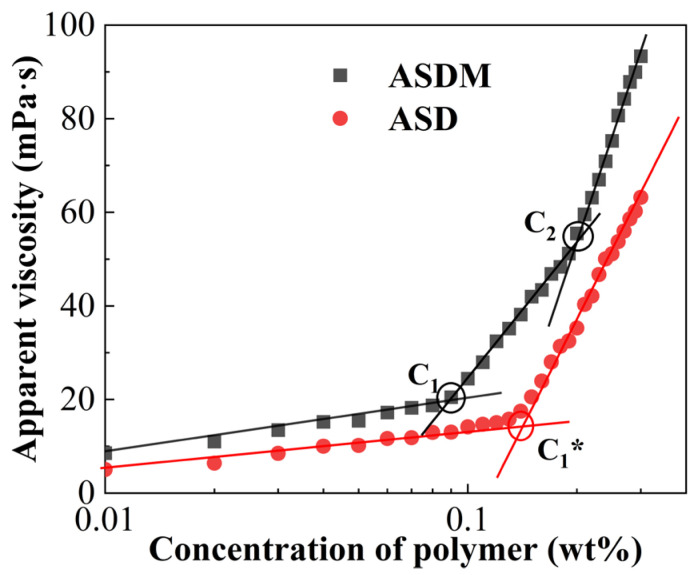
The CAC of ASD and ASDM.

**Figure 5 molecules-28-05104-f005:**
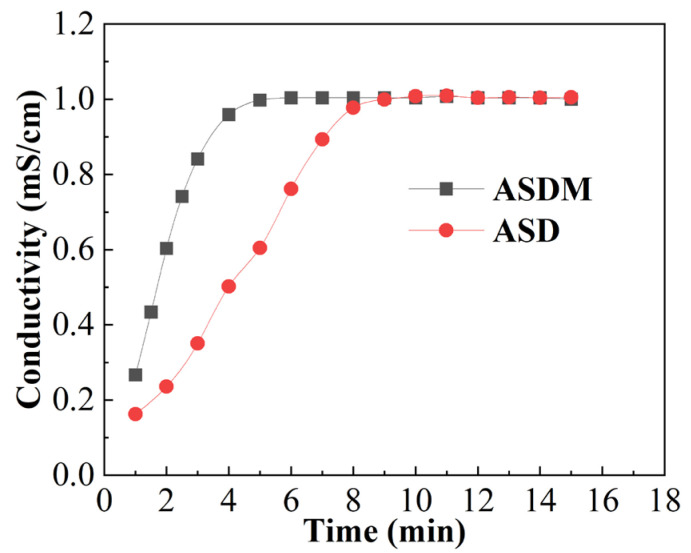
Changes in solution conductivity as a function of time during the dissolution of 0.2 wt% polymers.

**Figure 6 molecules-28-05104-f006:**
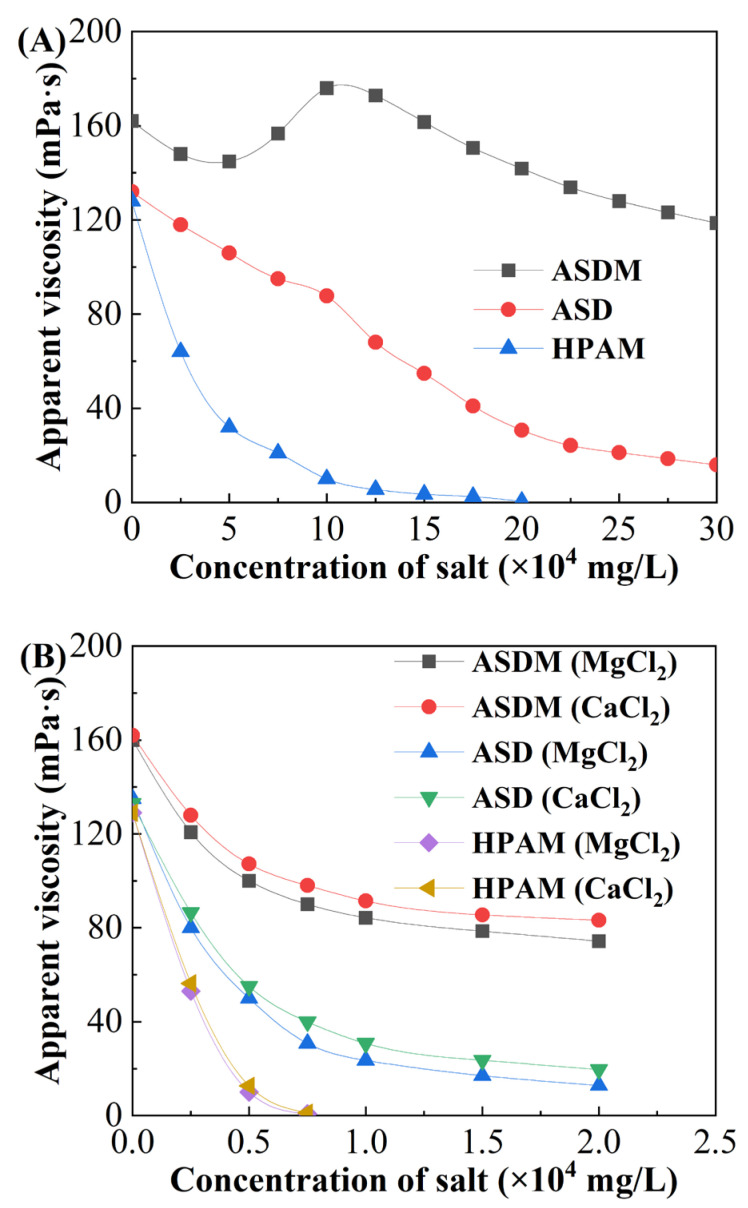
Variation in the viscosity as a function of inorganic salt concentrations. (**A**) Compound salt brine. (**B**) MgCl_2_ and CaCl_2_.

**Figure 7 molecules-28-05104-f007:**
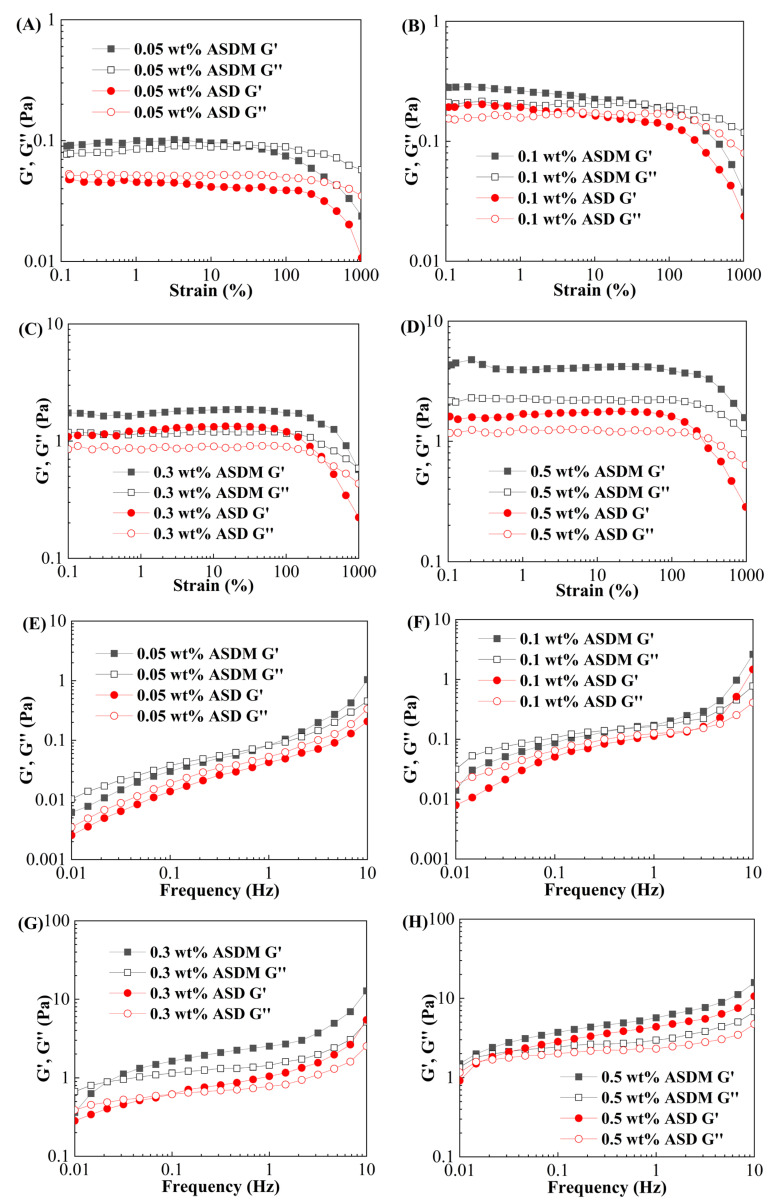
The Viscoelasticity of ASDM and ASD. (**A**–**D**) strain scanning curve, (**E**–**H**) frequency scanning curve.

**Figure 8 molecules-28-05104-f008:**
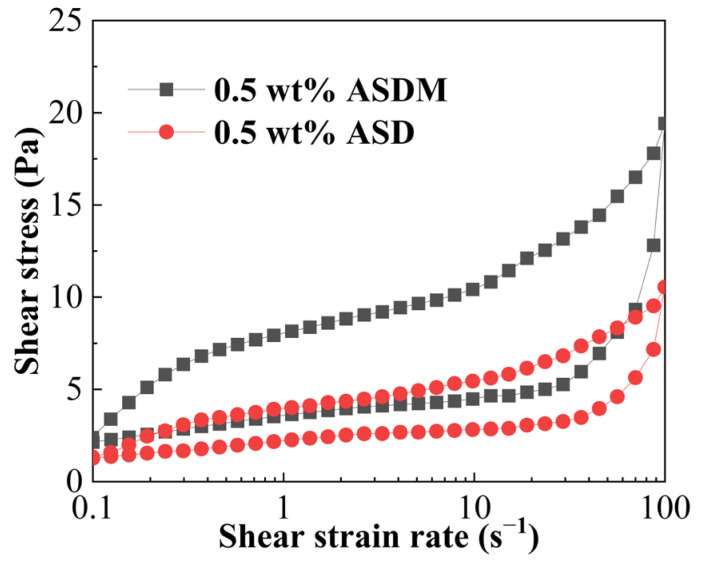
The thixotropic loop of the 0.5 wt% ASDM solution and 0.5 wt% ASD solution.

**Figure 9 molecules-28-05104-f009:**
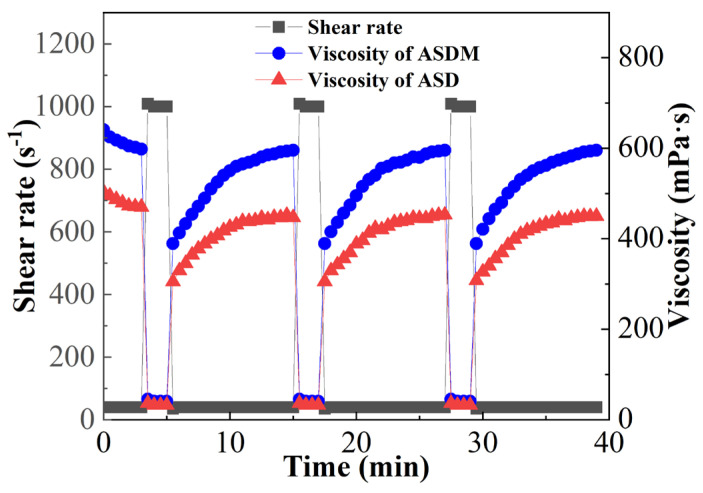
Shear recovery measurement of 0.5 wt% ASDM solution and 0.5 wt% ASD solution.

**Figure 10 molecules-28-05104-f010:**
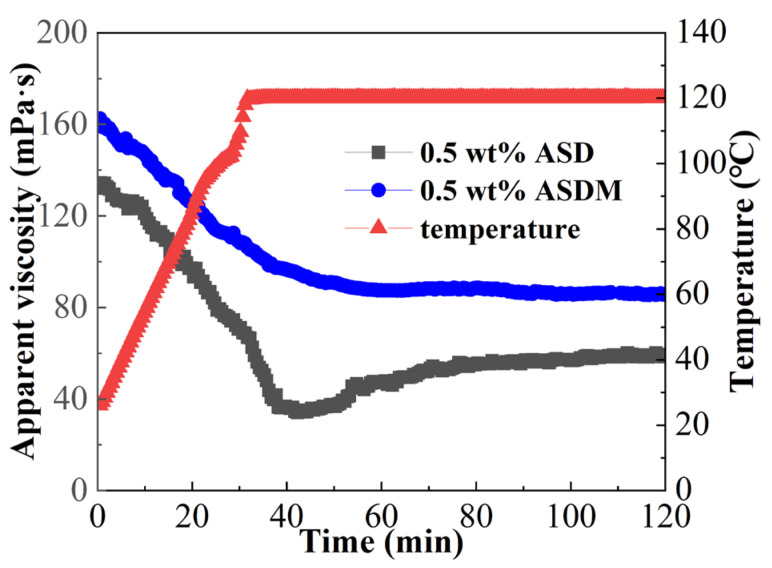
Temperature- and shear-resistance performance of 0.5 wt% ASDM solution and 0.5 wt% ASD solution.

**Figure 11 molecules-28-05104-f011:**
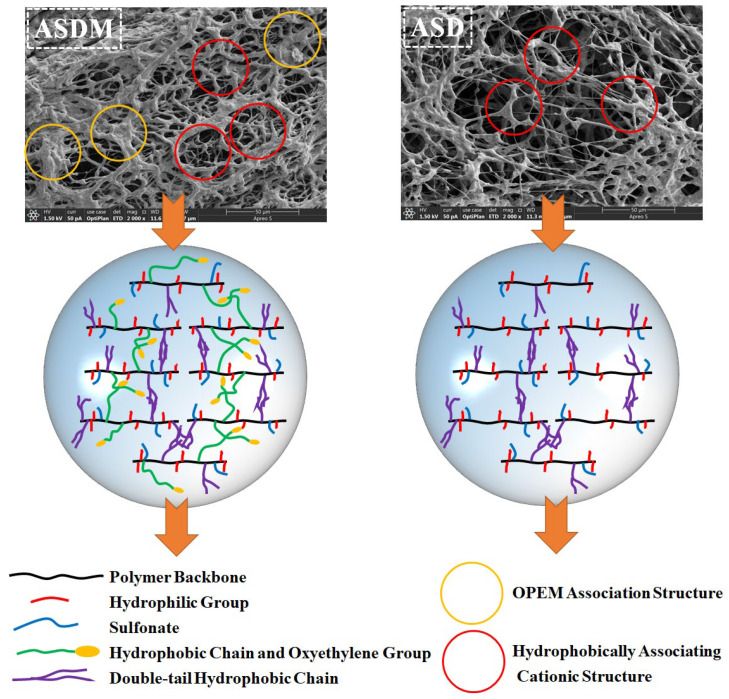
Morphological and mechanical images of ASD and ASDM.

**Figure 12 molecules-28-05104-f012:**
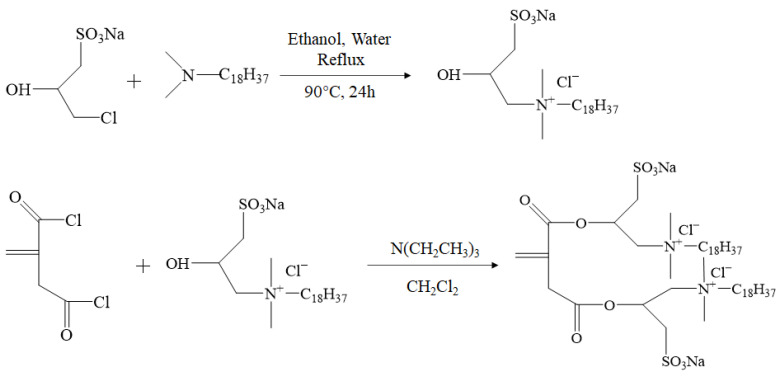
Synthesis route of double-tail hydrophobic monomer.

**Figure 13 molecules-28-05104-f013:**
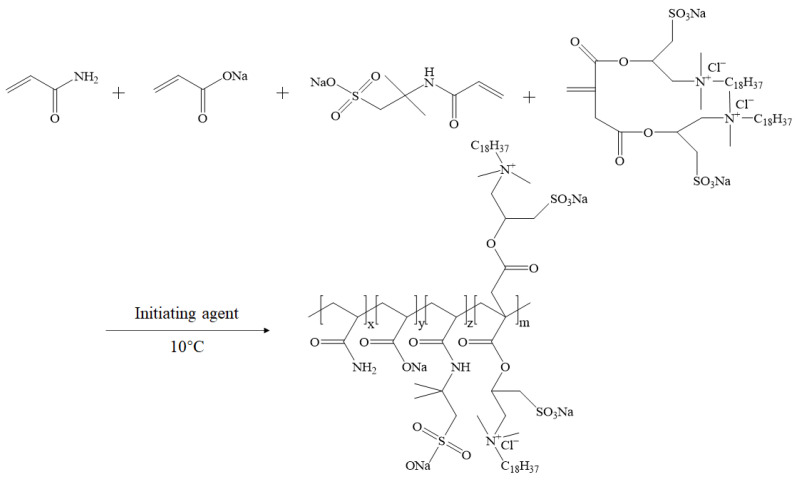
Synthesis route of ASD.

**Figure 14 molecules-28-05104-f014:**
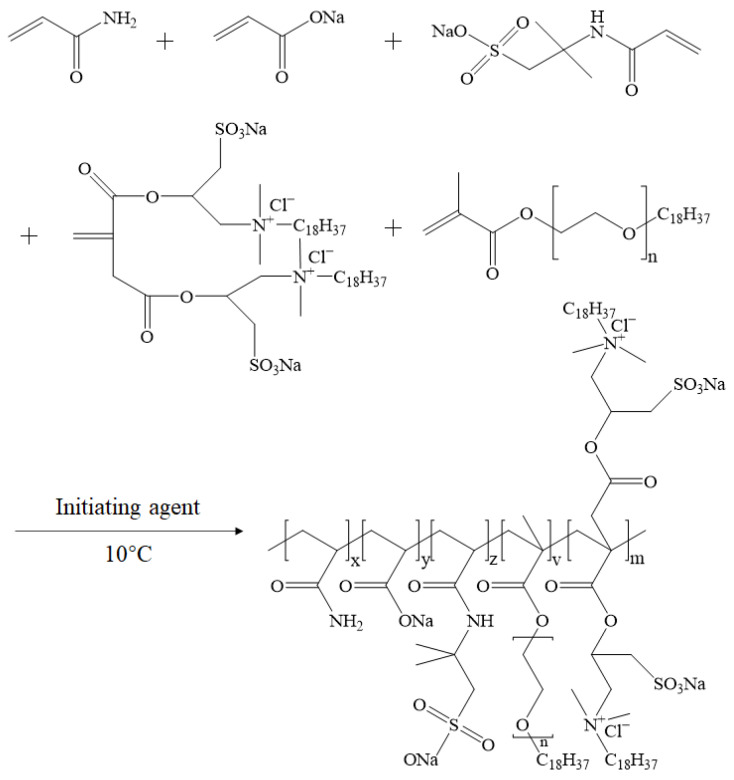
Synthesis route of ASDM.

**Table 1 molecules-28-05104-t001:** Test of proppant carrying capacity.

Fracturing Fluid	Temperature (°C)	Setting Velocity (mm/s)
0.3 wt% ASDM	25	0.028
0.3 wt% ASD	25	0.053
0.5 wt% ASDM	90	0.082
0.5 wt% ASD	90	0.147
0.8 wt% ASD	120	0.153
1.0 wt% ASD	140	0.228

## Data Availability

The data presented in this study are available on request from the corresponding author.

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
