# Peer review of "Fracturing Fluid Polymer Thickener with Superior Temperature, Salt and Shear Resistance Properties from the Synergistic Effect of Double-Tail Hydrophobic Monomer and Nonionic Polymerizable Surfactant"

_molecules, 2023, doi:10.3390/molecules28135104_

Round 1

Reviewer 1 Report

The manuscript is correctly written, scientifically sound, and clearly presented in good written English. Employed experimental methods are adequate to allow repetition of the work. Data are correctly interpreted to support the conclusions. Relevant issues in the discussion are adequately discussed.

Major concerns:

In this work, new hydrophobic-associating polymers as fracturing fluid thickeners were synthesized through aqueous solution polymerization and compared. Although the author reports a complete investigation, they need to clarify the below points.

-          Suggest several possible applications of described polymers, a detailed explanation linked with the structure is highly recommended.

-          Please add raw data of NMR with assignments.

-          If the authors have made some other characterization techniques they should present them in the manuscript.

- Please reduce the use of abbreviations throughout the manuscript, especially through the text of the abstract. Some sentences are very confusing and hard to understand.

- Compare these newly synthesized polymers with other polymers

Minor concerns:

The manuscript has some sp

Minor editing of the English language is required.

Author Response

Dear reviewer

I am very grateful to your comments for the manuscript. Those comments are valuable and very helpful. We have read through comments carefully and have made corrections. The responses to the reviewer's comments are marked in red and presented following.

  1. Suggest several possible applications of described polymers, a detailed explanation linked with the structure is highly recommended.

Response: Thank you for your suggestion.

The described polymer (ASDM) was synthesized by acrylamide (AM), acrylic acid (AA), 2-acrylamido- 2-methylpropane sulfonic acid (AMPS), nonionic polymerizable surfactant (NPS) and double-tail hydrophobic monomers (DHM).

Acrylamide based polymer is a major water-soluble polymer with the characteristics of flocculation, thickening, drag reduction and oil displacement. To achieve the effects of strong hydration, temperature resistance, salt resistance and shear resistance, in addition to selecting Acrylamide (AM) and/or acrylic acid (AA) as the monomers for the polymer main chain, the side chain design of synthetic polymers used in the field usually introduces several functional monomers at the same time rather than a single functional group.

The main chain of Polyacrylamide (PAM) is connected by C-C single bonds with high bond energy (340 kJ/mol). High thermal energy is required to destroy C-C bonds, which determines that PAM is not easy to degrade at high temperature. Although PAM does not contain glycosidic bonds mainly responsible for the thermal degradation of polysaccharides, the thermal confinement for PAM indeed exists. The structural integrity of the polymers will be destroyed through amide hydrolysis and other decomposition pathways at high temperature, resulting in lost viscosity. The nitrogen atoms of the amide groups or the oxygen atoms of the carboxyl groups formed by the hydrolysis of the amide groups, on the PAM molecular chains, can provide lone electron pairs as crosslinking points for complexation reaction to form a high temperature resistant crosslinked gel.

Anionic monomer 2-acrylamide-2-methylpropane sulfonate (AMPS) is the most commonly used rigid monomer for modified PAM. The steric effect of the huge side group (tertiary butyl) of AMPS and the strong ionization of the sulfonic acid group increases the rigidity of the macromolecular chains, making the sulfonate Acrylamide/AMPS copolymers excellent water-solubility, temperature and salt resistance.

A small amount of hydrophobic monomer (<2 mol%) is introduced to form hydrophobically associated polyacrylamide (HAPAM). Very similar to surfactant micelles, HAPAM in the aqueous solution changes from intramolecular hydrophobic group association to intermolecular hydrophobic group association at a concentration higher than the critical association concentration (CAC), forming a dynamic physical crosslinking network. Increasing the length or number of hydrophobic blocks is helpful to improve the temperature resistance of polymers, but the excessive hydrophobic block is not conducive to the dissolution of the product in water. Double-tailed hydrophobic monomers on the molecular chain are more conducive to the intermolecular association, greatly improving the rheological properties and temperature, shear and salt resistance of polymers than single-tailed hydrophobic monomers.

However, HAWSP prepared through micellar copolymerization usually have some disadvantages, the long dissolution time and slow dissolution rate of HAWSP, unreacted residual surfactants may lead to environmental problems, and the presence of surfactants in the final product may cause some adverse effects on product performance. The introduction of polymerizable surfactant can solve these problems in traditional surfactant applications. Polymerizable surfactant molecules incorporate polymerizable reactive groups, mainly unsaturated alkenes, into the structure of conventional surfactant molecules. In contrast to conventional surfactants, polymerizable surfactants not only possess the characteristics of conventional surfactants, but also can be copolymerized with unsaturated monomers at high temperature or in the presence of initiators, and these compounds can also be used to improve the solubility and association of HAWSP. Polymerizable surfactants can replace common hydrophobic monomers and surfactants and can be further locked into these new supramolecular structures by polymerization.

  1. Please add raw data of NMR with assignments.

Response: The nuclear magnetic resonance proton spectra (1H NMR) of the polymers in deuteri-um chloride (D2O) were measured using a Bruker AVANCE NEO 600 spectrometer (Bruker, Karlsruh, Germany), the concentration of the polymer solution was 100 mg/L.

Figure 3 shows the 1H NMR (400 MHz, D2O) spectrum of ASDM. It was shown from Figure 3 that the proton signal at 4.70 ppm was allocated to the solvent proton (D2O). 0.78 ppm(a, -CH3), 1.10-1.12 ppm(n, -CH3), 1.22 ppm(b, -CH2-CH3), 1.44 ppm(d, -CH3, AMPS), 1.56 ppm(m, -CH2-), 2.12 ppm(e, -CH-), 2.41 ppm(l, -CH2-COO-), 3.27 ppm(c, -CH2-CH2-), 3.55 ppm(g, -CH3), 3.57 ppm(h, -CHCH2-N-), 3.59 ppm(i, -CH-CH2-SO3), 3.60 ppm(k, -C-CH2-SO3, AMPS), 3.63 ppm(f, -O-CH2-), 5.56-5.59 ppm(j, -O-CH-). Therefore, 1H NMR analyses confirmed that the polymer produced in this study was largely consistent with the designed polymer, which indicated successful synthesis.

Figure 3. 1H NMR spectra of ASDM.

  1. If the authors have made some other characterization techniques they should present them in the manuscript.

Response: The synthesized polymer was characterized by Fourier transform infrared (FTIR) and 1H nuclear magnetic resonance (NMR).

  1. Please reduce the use of abbreviations throughout the manuscript, especially through the text of the abstract. Some sentences are very confusing and hard to understand.

Response: Abstract is revised as follows:

To develop high-salinity, high-temperature reservoirs, two hydrophobically associating polymers as fracturing fluid thickener were respectively synthesized through aqueous solution polymerization with acrylamide (AM), acrylic acid (AA), 2-acrylamido-2-methylpropanesulfonic acid (AMPS), nonionic polymerizable surfactant (NPS) and double-tail hydrophobic monomer (DHM). The thickener ASDM (AM/AA/AMPS/NPS/DHM) and thickener ASD (AM/AA/AMPS/DHM) were compared in terms of properties of water dissolution, thickening ability, rheological behavior and sand-carrying. The results showed that ASDM could be quickly diluted in water within 6 min, 66.7% less than that of ASD. ASDM exhibited salt thickening performance, the apparent viscosity of 0.5 wt% ASDM reached 175.9 mPa·s in 100,000 mg/L brine, 100.6% higher than that of ASD. The viscosity of 0.5 wt% ASDM was 85.9 mPa s after shearing for 120 min at 120 °C and at 170 s-1, 46.6% higher than that of ASD. ASDM exhibited better performance in thickening ability, viscoelasticity, shear recovery, thixotropy and sand-carrying. The synergistic effect of hydrophobic association and linear entanglement greatly enhancing the performance of ASDM and the compactness of the spatial network structure of the ASDM was enhanced. In general, ASDM exhibited great potential for application in extreme environmental conditions with high salt and high temperature.

  1. Compare these newly synthesized polymers with other polymers

Response: We have compared the salt resistance of newly synthesized polymers with HPAM.

The effects of salt on the viscosity of the polymer solution by measuring the viscosity at different concentrations of compound salt brine (NaCl and CaCl2 at a mass ratio of 10: 1), MgCl2 and CaCl2 are shown in Figure 2. As shown in Figure 2a, the viscosity of the 0.5 wt% ASD decreased as the salt concentration increased. At high salt concentration, hydrophobic monomers reduced the solubility of the polymer, resulting in a rapid decrease in its viscosity. The viscosity of 0.5 wt% ASD was below 90 mPa s when the salt concentration was greater than 10 × 104 mg/L, the viscosity of the ASDM solution was increased significantly with the introduction of NPS compared to ASD, the viscosity of 0.5 wt% ASDM solution increased from 162.1 mPa s to 175.9 mPa s when salt concentration reached 10 × 104 mg/L, 100.6 % higher than that of ASD, and the ASDM exhibited salt thickening. As the salt concentration increased further, the viscosity of the ASDM solution decreased gradually, and finally decreased to a stable value of about 118.6 mPa·s. When the concentration of compound salt reached 200,000 mg/L, the viscosity of HPAM solution decreased almost to zero.

The polymer chains usually undergo curling and viscosity loss due to the electric double layer compression of the polymer hydration shell and electrostatic shielding effect induced by metal ions. For hydrophobically associating polymer ASDM, AMPS was widely used as a salt-tolerant functional group in polymer design because of its strong anionic and water-soluble sulfonic acid group, which was not sensitive to the attack by external ions. The introduction of oxyethylene groups in NPS with strong hydration capacity improved the solubility and inhibited hydrolysis of the ASDM at high brine concentrations. In addition, the hydrophobic association effect between NPS and DHM was enhanced due to solution polarity enhancement, the electric double layer was compressed by salt ions, which facilitated the association of hydrophobic chains. Complexation reactions would be occurred between multivalent metal ions and oxyethylene groups by filling the unoccupied orbital of metal ion through the lone pair of electrons from the oxygen of the oxyethylene groups. As a result, the repulsion between hydrophobic groups and oxyethylene groups was enhanced, subsequently promoting the stretching of the polymer chains. All of the above factors increased the hydrodynamic volume of polymers and the viscosity of the polymer solution, contributing to the resistance of inorganic mineral salts. Therefore, an increase in salt concentration had a positive effect on the viscosity of the ASDM solution. Subsequently, more inorganic salts were added to the polymer solution, and the excess inorganic salt ions dehydrated the polymer molecules, the electrostatic shielding effect of salt on polymer ASDM molecular chains exceeded the hydrophobic association effect between polymer molecules, leading to a thinning of the hydration shells of the polymer molecules. Finally, the viscosity of ASDM solution decreased at high salt concentration. Meanwhile, the carboxylate ions on the HPAM molecule were shielded by the metal cation in the brine, causing the polymer chains to curl, the viscosity of the HPAM solution decreased sharply.

The effect of MgCl2 and CaCl2 on the viscosities of polymer solutions were also investigated, and the results were shown in Figure 2b. With the increase in the MgCl2 and CaCl2 solution concentration, the viscosities of the polymer solutions decreased. Obviously, the viscosity of ASDM solution was greater than ASD solution with increased of MgCl2 and CaCl2, indicating that there was a synergistic effect between NPS and DHM. When the concentrations of MgCl2 and CaCl2 reached 2×104 mg/L, the viscosities of ASDM solutions decreased from 106.5 mPa s to 74.3 mPa s and 83.2 mPa s, respectively. However, the viscosity of HPAM solution was almost zero when the concentrations of calcium chloride and magnesium chloride reached 7,000 mg/L, indicating that the polymer ASDM obtained excellent salt tolerance compared with HPAM. In addition, it could be seen from the Figure 5 that the three metal ions of Na+, Ca2+, and Mg2+ had different effects on the viscosity of the polymer solution, which was caused by the different hydration ionic radiuses of the three mental ions. The larger the radius of hydrated ions, the greater the attraction between the salt ions and the water molecules, and the stronger dehydration ability of inorganic salt ions. From a macroscopic point of view, the salt ions (Ca2+ and Mg2+) with a large hydrated ionic radius caused the viscosity of the polymer solution to decreased sharply to be stable. The influence mechanism of divalent ions was the same as that of monovalent ions, but the compression of the electric double layer of the polymer hydration shell was strengthened.

Figure 2. Variation in the viscosity as a function of inorganic salt concentrations, (a) compound salt brine, (b) MgCl2 and CaCl2.

  1. The manuscript has some sp

Response: I will correct spelling mistakes carefully one by one.

Reviewer 2 Report

The manuscript by S. Shi et al reports on their recent findings of viscoelastic behavior and morphology of copolymers composed of hydrophilic and hydrophobic monomeric units proposed as thickeners for petroleum industry. They show that introduction of both ionizable groups (acrylic and 2-acrylamido-2-methylpropanesulfonic acid) and hydrophobic double-tail monomers allows improvement in thickening ability, thixotropy, and complex viscoelastic behavior of their solutions due to formation of spatial network structures and hydrophobic association. The work may be of interest to a number of researchers working with polymers far beyond petroleum engineers.

Only few flaws of “cosmetic” character should be fixed, in my opinion.

1.      First is terminology used by the authors. The term “associative polymers” is very rarely used in the literature: I found only one instance: the paper by Mitchell Winnik and Ahmad Yekta, 1997 (https://doi.org/10.1016/S1359-0294(97)80088-X). By the way, I recommend citing this paper as one of pioneering accounts on this matter. Most researchers use term “associating polymers”.

2.      Second is the title. I recommend to add a word “polymers” or “macromolecules” both to the title and keywords to make instant reference to the nature of the “thickeners”.

3.      It is difficult to follow the abbreviations used in the paper (OPEM, MAOZ2). I recommend mnemonic abbreviations.

4.      A reader will appreciate the scheme where all monomers’ structures and chemical composition of the polymers introduced in the beginning of the paper, may be in the intro.

5.      Beyond these comments, I would like the authors to discuss in greater details the role of ionizing groups in the context of associating polymeric thickeners (Figure 4). Which salt was used in this case? What is the role of ionizing groups in this case? How would the presence of di- and trivalent cations (Ca+2, Mg+2, Al+3) affect the behavior of the ASD and ASDM in real life application?

I may recommend this manuscript for publication once all issues are properly addressed in the text of the manuscript.

Author Response

Dear reviewer

I am very grateful to your comments for the manuscript. Those comments are valuable and very helpful. We have read through comments carefully and have made corrections. The responses to the reviewer's comments are marked in red and presented following.

  1. First is terminology used by the authors. The term “associative polymers” is very rarely used in the literature: I found only one instance: the paper by Mitchell Winnik and Ahmad Yekta, 1997 (https://doi.org/10.1016/S1359-0294(97)80088-X). By the way, I recommend citing this paper as one of pioneering accounts on this matter. Most researchers use term “associating polymers”.

Response: Thank you for your suggestion. The “associative polymers” in the manuscript have all been modified to“associating polymers”.

  1. Second is the title. I recommend to add a word “polymers” or “macromolecules” both to the title and keywords to make instant reference to the nature of the “thickeners”.

Response: The title has been revised to Fracturing Fluid Polymer Thickener with Superior Temperature, Salt and Shear Resistance Properties from The Synergistic Effect of Double-tail Hydrophobic Monomer and Nonionic Polymerizable Surfactant.

Keywords: fracturing fluid polymer thickener; high-salinity and high-temperature; hydrophobic association; linear entanglement; synergistic effect

  1. It is difficult to follow the abbreviations used in the paper (OPEM, MAOZ2). I recommend mnemonic abbreviations.

Response: OPEM has been revised to NPS, NPS is the first letter of the words nonionic polymerizable surfactant. D-MAOZ2 has been revised to DHM, which stands for the first letter of the words double-tail hydrophobic monomer.

  1. A reader will appreciate the scheme where all monomers’ structures and chemical composition of the polymers introduced in the beginning of the paper, may be in the intro.

Response: Thank you for your suggestion. all monomers’ structures and chemical composition of the polymer ASDM introduced in the “Introduction” of the manuscript, shown in Figure 1.

Figure 1. all monomers’ structures and chemical composition of the polymer ASDM. (a) monomers’ structure, (b) polymer ASDM, (c) polymer ASD.

  1. Beyond these comments, I would like the authors to discuss in greater details the role of ionizing groups in the context of associating polymeric thickeners (Figure 4). Which salt was used in this case? What is the role of ionizing groups in this case? How would the presence of di- and trivalent cations (Ca+2, Mg+2, Al+3) affect the behavior of the ASD and ASDM in real life application?

Response:

It is well known that polymers are often dissolved in brine composed of several inorganic salts from an application point of view. Hence, after evaluating the resistance of polymers to a single inorganic salt, it was necessary to evaluate the effect of compound salts (NaCl and CaCl2 at a mass ratio of 10: 1) on the viscosities.

The effects of salt on the viscosity of the polymer solution by measuring the viscosity at different concentrations of compound salt brine (NaCl and CaCl2 at a mass ratio of 10: 1), MgCl2 and CaCl2 are shown in Figure 2. As shown in Figure 2a, the viscosity of the 0.5 wt% ASD decreased as the salt concentration increased. At high salt concentration, hydrophobic monomers reduced the solubility of the polymer, resulting in a rapid decrease in its viscosity. The viscosity of 0.5 wt% ASD was below 90 mPa s when the salt concentration was greater than 10 × 104 mg/L, the viscosity of the ASDM solution was increased significantly with the introduction of NPS compared to ASD, the viscosity of 0.5 wt% ASDM solution increased from 162.1 mPa s to 175.9 mPa s when salt concentration reached 10 × 104 mg/L, 100.6 % higher than that of ASD, and the ASDM exhibited salt thickening. As the salt concentration increased further, the viscosity of the ASDM solution decreased gradually, and finally decreased to a stable value of about 118.6 mPa·s. When the concentration of compound salt reached 200,000 mg/L, the viscosity of HPAM solution decreased almost to zero.

The polymer chains usually undergo curling and viscosity loss due to the electric double layer compression of the polymer hydration shell and electrostatic shielding effect induced by metal ions. For hydrophobically associating polymer ASDM, AMPS was widely used as a salt-tolerant functional group in polymer design because of its strong anionic and water-soluble sulfonic acid group, which was not sensitive to the attack by external ions. The introduction of oxyethylene groups in NPS with strong hydration capacity improved the solubility and inhibited hydrolysis of the ASDM at high brine concentrations. In addition, the hydrophobic association effect between NPS and DHM was enhanced due to solution polarity enhancement, the electric double layer was compressed by salt ions, which facilitated the association of hydrophobic chains. Complexation reactions would be occurred between multivalent metal ions and oxyethylene groups by filling the unoccupied orbital of metal ion through the lone pair of electrons from the oxygen of the oxyethylene groups. As a result, the repulsion between hydrophobic groups and oxyethylene groups was enhanced, subsequently promoting the stretching of the polymer chains. All of the above factors increased the hydrodynamic volume of polymers and the viscosity of the polymer solution, contributing to the resistance of inorganic mineral salts. Therefore, an increase in salt concentration had a positive effect on the viscosity of the ASDM solution. Subsequently, more inorganic salts were added to the polymer solution, and the excess inorganic salt ions dehydrated the polymer molecules, the electrostatic shielding effect of salt on polymer ASDM molecular chains exceeded the hydrophobic association effect between polymer molecules, leading to a thinning of the hydration shells of the polymer molecules. Finally, the viscosity of ASDM solution decreased at high salt concentration. Meanwhile, the carboxylate ions on the HPAM molecule were shielded by the metal cation in the brine, causing the polymer chains to curl, the viscosity of the HPAM solution decreased sharply.

The effect of MgCl2 and CaCl2 on the viscosities of polymer solutions were also investigated, and the results were shown in Figure 2b. With the increase in the MgCl2 and CaCl2 solution concentration, the viscosities of the polymer solutions decreased. Obviously, the viscosity of ASDM solution was greater than ASD solution with increased of MgCl2 and CaCl2, indicating that there was a synergistic effect between NPS and DHM. When the concentrations of MgCl2 and CaCl2 reached 2×104 mg/L, the viscosities of ASDM solutions decreased from 106.5 mPa s to 74.3 mPa s and 83.2 mPa s, respectively. However, the viscosity of HPAM solution was almost zero when the concentrations of calcium chloride and magnesium chloride reached 7,000 mg/L, indicating that the polymer ASDM obtained excellent salt tolerance compared with HPAM. In addition, it could be seen from the Figure 5 that the three metal ions of Na+, Ca2+, and Mg2+ had different effects on the viscosity of the polymer solution, which was caused by the different hydration ionic radiuses of the three mental ions. The larger the radius of hydrated ions, the greater the attraction between the salt ions and the water molecules, and the stronger dehydration ability of inorganic salt ions. From a macroscopic point of view, the salt ions (Ca2+ and Mg2+) with a large hydrated ionic radius caused the viscosity of the polymer solution to decreased sharply to be stable. The influence mechanism of divalent ions was the same as that of monovalent ions, but the compression of the electric double layer of the polymer hydration shell was strengthened.

Figure 2. Variation in the viscosity as a function of inorganic salt concentrations, (a) compound salt brine, (b) MgCl2 and CaCl2.

Round 2

Reviewer 1 Report

Accept in present form.

Minor editing of English language required.